# Semi-Autonomous Planning and Visualization in Virtual Reality

Gregory LeMasurier
University of Massachusetts Lowell
gregory_lemasurier@student.uml.edu

Jordan Allspaw
University of Massachusetts Lowell
jallspaw@cs.uml.edu

Holly A. Yanco
University of Massachusetts Lowell
holly@cs.uml.edu

## ABSTRACT

Virtual reality (VR) interfaces for robots provide a three-dimensional (3D) view of the robot in its environment, which allows people to better plan complex robot movements in tight or cluttered spaces. In our prior work, we created a VR interface to allow for the teleoperation of a humanoid robot. As detailed in this paper, we have now focused on a human-in-the-loop planner where the operator can send higher level manipulation and navigation goals in VR through functional waypoints, visualize the results of a robot planner in the 3D virtual space, and then deny, alter or confirm the plan to send to the robot. In addition, we have adapted our interface to also work for a mobile manipulation robot in addition to the humanoid robot. For a video demonstration please see the accompanying video at https://youtu.be/wEHZug_fxrA.

## CCS CONCEPTS

• **Human-centered computing** → **Virtual reality**; • **Computer systems organization** → **Robotic control**; *External interfaces for robotics.*

## KEYWORDS

Human-robot interaction (HRI), virtual reality (VR), shared control, manipulation planning, motion planning, Functional Waypoints

## 1 INTRODUCTION

The availability of Augmented Reality (AR) and Virtual Reality (VR) is leading many researchers to investigate different potential use cases of this technology, ranging from various training applications [3] to robot interfaces [19]. Using VR for robots is particularly appealing as three dimensional (3D) sensors are increasingly being used on robot systems to provide point clouds of their environment — and these point clouds can be visualized within the VR environment to provide the operator with an understanding of the robot's operating environment. Providing an operator with a 3D view of the environment that can be visualized from all directions improves situation awareness and can prevent robot collisions with the environment.

Our initial inspiration for using VR as a robot interface stems from our analysis of the DARPA Robotics Challenge (DRC) Finals [14], a competition where teams teleoperated a humanoid robot to perform several difficult tasks such as navigating through uneven terrain and operating a hand drill. From the analysis, we found that teams used a variety of different control methods, but autonomy was very limited. Many teams found that combining some level of direct joint control with specific scripted actions was necessary. For example, Team THOR used "sets of algorithms that gracefully switch among high level autonomous behaviors and low levels of teleoperated control" [12]. Team IHMC initially used teleoperated interfaces to send the robot into a known state, whereby an autonomous script could finish the task, but knew that such

an approach was brittle, and so transitioned to more interactive tools whereby teleoperation and autonomy could be combined to provide an easy and effective control [7]. Both approaches took advantage of human-in-the-loop planning, where a robot creates a plan based on sensors and human input, then human views the plan and potentially adjusts it, before allowing the robot to carry out the plan.

In our prior work, we set out to create a VR interface [1] with the goal of enabling an operator to complete a variety of tasks remotely, including robot navigation and various dexterous manipulation tasks. After internal testing of our previous interface, we found that, while functional, our prior design was limited to imitating many of the controls as they appeared in a traditional 2D interface (e.g., relying on similar menu options that are found in traditional 2D interfaces) in virtual 2D panels. We redesigned the interface to expand the controls to take advantage of the unique features in VR. In particular, our goal setting and plan visualization methods needed to be expanded to address limitations in our previous interface. We determined that new methods needed to be added for goal interactions to take advantage of the 3D virtual space and ultimately reduce the time an operator spends interacting with a menu interface. Additionally, our interface needed to be expanded to include a human-in-the-loop planning system.

To facilitate planning in a VR environment, we propose the concept of **Functional Waypoints**. Functional waypoints differ from traditional waypoints, in that they not only provide intermediate goals for the desired trajectory but also enable the operator to send additional commands to alter the robot's state at these intermediate points. Functional waypoints consist of a target goal position as well as state information that is used to provide additional control of the robot such as opening or closing the robot's endeffector, adjusting the height of the robot, changing the direction that a robot is looking, or specifying other joint states. Functional waypoints differ from Affordance Templates [6] as functional waypoints enable a higher resolution of joint control and a variety of joints to be modified. Additionally functional waypoints enable an operator to specify additional state information, such as collision avoidance, at each waypoint. By providing the operator with the ability to set additional functionality at each waypoint, we reduce the need for an operator to switch between control methods or user interface menus. This ultimately enables the operator to fluently create a complex plan for the robot to complete a task, meaning that after the plan is approved, the operator does not need to provide additional commands, freeing the operator to monitor the robot or to work on other tasks.

In analysis of our initial interface, we found that many actions required operators to use a 2D menu interface in the virtual world or to switch modes to send complex plans to the robot. This is not a fluent control method, as an operator would have to plan and execute a partial trajectory to each location where they wanted to send

additional commands to the robot. For example, in a pick and place task, functional waypoints can be used to send a trajectory with additional end effector state information that enables the operator to specify how open or closed the end effector should be at each waypoint. Without functional waypoints, an operator would need to use several control methods and send several plans in order to complete the same task. Our interface adds two types of functional waypoints, one for manipulation and another for navigation, to enable the operator to fluently create complex manipulation and navigation plans.

In this paper, we discuss improvements to the visualization and control scheme for our updated VR interface. In particular, we focus on the development of VR controls to allow an operator to create functional waypoints for a robot planner, and then visualize the resulting plan from the robot. At that point, the operator can approve, disprove, or adjust the plan as necessary. We have also generalized much of the interface to work with additional robots, including a wheeled mobile manipulation robot. Our goal is to allow an operator to control a robot to perform dexterous tasks entirely from within VR without ever needing to remove the headset in order to use a command line or a supplementary interface. We also want to allow the operator to complete the tasks quickly and, most importantly, accurately.

## 2 SYSTEM AND ENVIRONMENT

For our VR Headset we used an HTC Vive [13] with the two included controllers. We also have an alternative setup with the same headset, but with Manus VR Gloves [11] substituting the controllers. In both cases, there is position and orientation tracking of the operator's hands and head. The controllers provide the tracking natively while the Manus gloves are augmented with SteamVR Trackers that provide the feature. While the gloves do not have physical buttons, as provided on the controller, they add in accurate finger tracking and gesture control. In the work described in this paper, we have focused on the use of the controllers.

The HTC Vive headset is running on Windows using Unity[1] and ROS.NET[2] to communicate with the robot. We have also used a couple scripts from the VRTK[3] library, namely for teleportation and user interface (UI) interaction. The user interface components are modular, which allowed us to add ROS subscribers for common robot displays such as the robot description, camera streams, and point clouds. This modular design allowed us to easily adapt the VR interface to control different kinds of robots.

The first robot that we used for the VR interface development was the NASA Valkyrie R5 [15], a humanoid robot with two 7-DOF arms, each with a four fingered hand, in addition to its 3-DOF torso and 3-DOF neck. Valkyrie has an RGB-D sensor and a LiDAR in the head, along with two RGB cameras in the torso, setup in a stereo configuration. We expanded our platform options to include the Fetch Mobile Manipulation robot [18], a mobile robot which has a single 7-DOF arm. Fetch comes with a LiDAR located just above ground level for obstacle detection and avoidance during navigation, as well as a RGB-D camera in the head used for vision.

[1]https://unity.com/
[2]https://github.com/uml-robotics/ROS.NET_Unity
[3]https://vrtoolkit.readme.io/

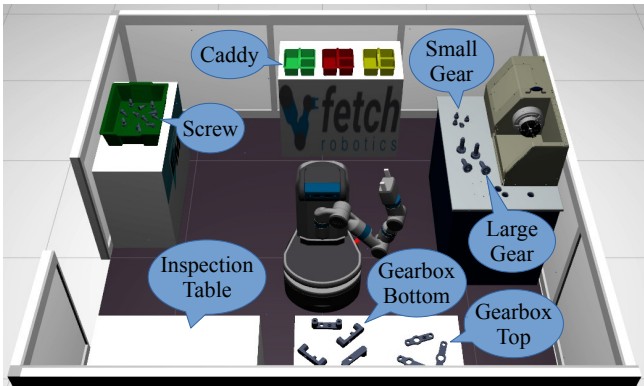

**Figure 1: Simulation of Fetch inside the "FetchIt! The Mobile Manipulation Challenge" Arena**

Both robots run ROS and thus are able to communicate with our interface. Fetch is running many standard packages including Moveit [4] for manipulation and the ROS Navigation stack [5] for SLAM and autonomous navigation. For Valkyrie, the manipulation and balancing code utilizes IHMC's [9] planner. Both robots have simulators in Gazebo [8], which we use for testing. In this paper, we used the Fetch robot for all examples.

For the Fetch robot, we used the arena seen in Figure 1 from the standardized task designed for the "FetchIt! The Mobile Manipulation Challenge" (at ICRA2019)[4] competition, where an autonomous Fetch robot was tasked with gathering objects from tables located around the robot in the arena, then placing the objects into specific compartments in a caddy. The objects were different sizes and in different conditions (i.e., gears and gearbox parts were spread out on a table, while screws were bunched up inside a bin). The completed caddy also needed to be picked up and placed at a designated location. In the challenge, the task needed to be completed fully autonomously; however, the task also serves as a good test environment for a operator interface. This environment is identical in both real world and simulation, with only some minor differences due to Gazebo's physics quirks. In the Fetchit challenge arena Fetch is surrounded by five tables: three of the tables contain objects the robot must grab, the top table has the caddies that the robot must deposit the objects into, and the inspection table is the dropoff location for the completed caddy.

Many other VR interfaces focus on efficiently completing a specific task, such as commanding a Rethink Robotics Baxter to pick and place objects on a table in front of the robot [10] or mobile robots navigating a maze [2]. This test environment allowed us to combine navigation and manipulation planning, while also requiring the operator to have good situation awareness of the remote environment to complete all components successfully without colliding with the environment.

## 3 MANIPULATION PLANNING IN VR

One major limitation of our prior interface was a lack of sophisticated control methods for robot manipulation. The operator can

[4]https://opensource.fetchrobotics.com/competition

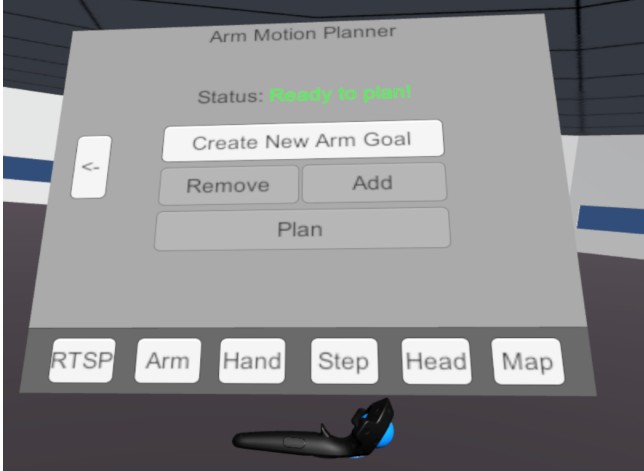

Figure 2: Virtual wristwatch user interface which the operator can activate by looking at their wrist, as if they were looking at a wristwatch. The operator can switch to different tabs using the buttons at the bottom of the interface.

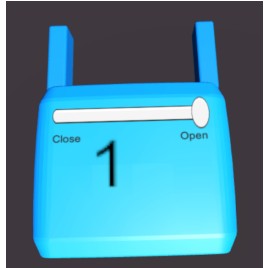

Figure 3: A manipulation functional waypoint for the Fetch robot, including a Fetch gripper model, a numeric label indicating the waypoint's order in the trajectory, and a slider interface to control how open or closed the gripper should be after the arm moves to this waypoint.

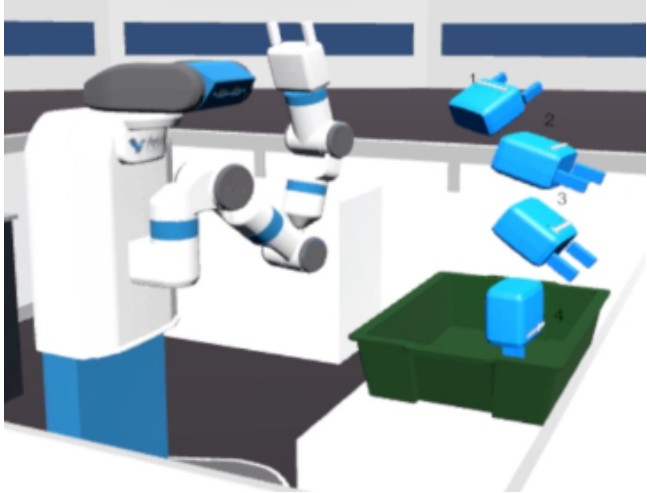

Figure 4: An example of an operator specified trajectory to command the Fetch robot to grab an object in the green bin using four manipulation functional waypoints.

only control the positioning of the robot's end effector in a very low level manner, through the use of sliders. In our previous paper, we described a virtual wristwatch user interface whereby the operator could look at their wrist as if it had a watch on it, which would open up a user interface attached to the controller as seen in Figure 2. We allowed the operator to switch between different tabs on the wristwatch depending on what the operator needed to do. One of the tabs included sliders that allowed the operator to individually control each joint between its entire range of motion. Sliders are beneficial to make adjustments, but they are very difficult to use to command a robot as they require the adjustment of one joint at a time. It would be very difficult for a novice user to understand how a robot's joints would need to move to complete their desired trajectory.

We had also provided a very high level direct control method. This method would enable a novice user to easily guide the robot through a trajectory, by moving their own arms through their desired path, then having the robot mimic the trajectory using inverse kinematics. Since this method occurs in real time the operator has no way to view the robot plan before it occurs. It is also difficult to interact with other VR elements without disabling direct control. To do a simple action such as pick up an object, the operator would need to enable direct control, guide the robot towards an object, disable direct control, close the gripper, enable direct control again, and guide the robot away. Switching modes like this is less than ideal for the operator as it over complicates the process. While very fast, this method is also problematic because the inverse kinematics solver can sometimes produce undesirable solutions, such as irregular joint configurations, or needlessly move uncomfortably close to an obstacle. Since the method is instantaneous, the operator has no opportunity to correct a problem before it occurs. In many cases it can be more desirable to have a human-in-the-loop planning, such as those used in the DRC interfaces discussed previously, whereby

the operator can view, alter, and approve the plan proposed by the robot.

Thus, we expand upon our previous interface by adding a new set of functional waypoint style virtual artifacts [17], or elements that can be moved around the virtual world by the operator, to create and view manipulation plans. Using the manipulation functional waypoint artifacts, as shown in Figure 3, the operator can quickly create and view a series of goal states for the robot's motion planner. In this case, Moveit will then plan the trajectory and return the completed plan, which the operator can then view in the virtual world. An example of this can be seen in Figure 4, where you can see the four solid blue manipulation functional waypoints created by the operator. After the functional waypoints are sent to the planner, the system displays the white and blue virtual robot mirroring its real world position, and the turquoise robot currently demonstrating the completed plan, as seen in Figure 5. If the operator is satisfied with the completed plan, they can then confirm it and watch the robot execute its plan. This approach intends to preserve and take

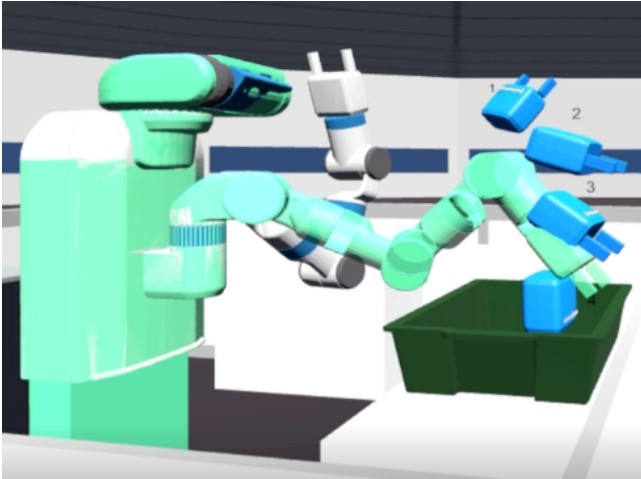

**Figure 5: The turquoise virtual copy of the robot shows the robot's planned trajectory to the operator.**

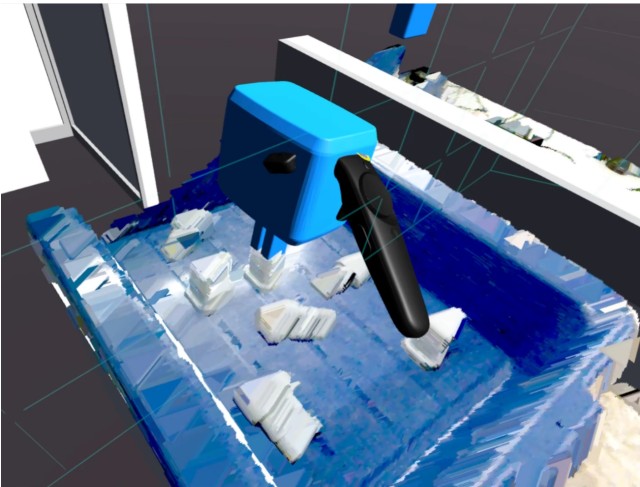

**Figure 6: Operator aligning the manipulation functional waypoint with the desired object.**

advantage of the operator's situational awareness by keeping them engaged with the robot environment they are working in, rather than dealing with the tabs in the wristwatch user interface.

The manipulation functional waypoints use a 3D model of the robot's end effector. The model is copied from the virtual robot which is created using the ROS robot description parameter, which only requires specifying the link name of the end effector when changing robots, allowing the implementation to be fairly robot agnostic. By using a 3D model of the robot's end effector, the operator can specify the position and orientation of goals for the robot's end effector with a complete spatial understanding of the goal they are sending to the robot. In combination with the point cloud visualization, the operator can ensure that their functional waypoints are not colliding with obstacles. In addition, the operator is able to identify a target for a pick action by looking at the point cloud visualization. Then they can place a manipulation functional waypoint in the appropriate grasping position for this object.

Additionally, these goals are labeled with a number above the gripper. This number indicates the order that the functional waypoints will be executed, to prevent any confusion to the operator. These labels also serve as an identifier, so that our interface can provide more meaningful feedback of the motion plan and during execution of the plan. This label will always will face the operator so that it is readable at all times.

Finally, the manipulation functional waypoints have a slider interface that enables the operator to specify how open or closed the gripper should be after executing the motion plan to this functional waypoint. This feature is what makes these functional waypoints rather than traditional waypoints, as additional commands are sent using the state information from the slider interface. This is particularly useful when an operator wants to grasp an object, for example. An example of this interface can be seen in Figure 7. As this is a slider interface, the operator can specify how closed the robot's gripper should be. The fingers in the manipulation functional waypoint model will move according to the slider value, enabling the operator to be aware of how the robot's fingers align to

the point cloud of the desired object as seen in Figure 6. The operator can thus specify a trajectory where the robot can pick up or release an object, without requiring the operator to switch between modes or open a tab in the wristwatch user interface. When executing the motion plan, the robot will first execute the trajectory to reach the first functional waypoint, then it will execute the end effector command, if the operator moved the slider. After the end effector command is completed, the robot will then continue to the next functional waypoint, if any, in the operator's specified trajectory.

## 3.1 Waypoint Creation and Modification

We further expand our control methods to enable operators to create, remove, and modify the manipulation functional waypoints. To take advantage of situational awareness in VR, the operator should have options to add functional waypoints without needing to open up the wristwatch interface. This enables the operator to maintain an understanding of the surrounding environment without losing focus while interacting with the tabs in the wristwatch user interface.

We offer three methods for creating new manipulation functional waypoints. The first is to simply grab onto the virtual robot's gripper which will spawn a new manipulation functional waypoint, at the end of the trajectory, in the operator's hand. The operator can then move this manipulation functional waypoint touching it with their controller and pressing and holding the grip button located on the side of the controller. While holding the grip button, the manipulation functional waypoint will follow the operator's virtual controller, much like they were holding the object. When the manipulation functional waypoint is in the desired location, the operator can release the grip button which will stop the object from following the controller, as if they had let go of the object.

The second way of creating a manipulation functional waypoint is to press the trigger button, typically where the index finger holds the controller, while holding a manipulation functional waypoint. This leaves the current functional waypoint where it was when the

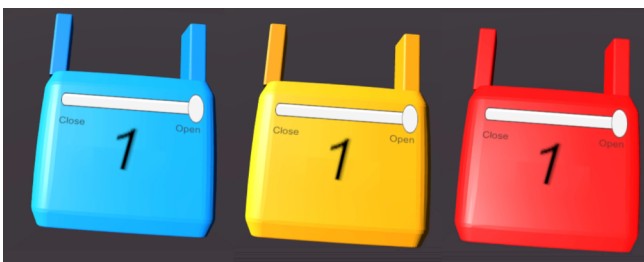

**Figure 7: Fetch manipulation functional waypoints in the default state (left, blue), pre-check determined to be out of reach from the robot (middle, orange), and planner error (right, red)**

operator pressed the button, and creates a new functional waypoint at the controller's current location. The newly created functional waypoint is always placed after the functional waypoint the operator was holding. For example, if there were already four functional waypoints, and the operator grabbed the first functional waypoint, labeled with a number one, and duplicated it, their new functional waypoint would be second in the trajectory, and the previous functional waypoints two, three and four would adjust themselves to be three, four, and five.

Both of these methods allow for a simple way of creating functional waypoints without switching between different modes, ultimately creating a more fluid way of setting a trajectory for the robot to follow while maintaining situational awareness. However both of these methods require the operator to be near the virtual robot, or one of their goals, which may not be possible or desirable in all situations. For example, if the robot has a very long arm that the operator cannot reach, or if the operator is on the other side of the virtual room but still wishes to create goals. To handle these situations, we have also created a third method of creating manipulation functional waypoints using the wristwatch user interface. By opening the wristwatch user interface, selecting the manipulation tab, and pressing the create waypoint button the operator can create a new functional waypoint a short distance in front of them, near eye level. While the first two methods are much faster and more convenient, this method allows the operator to always be able to create functional waypoints even when they are not near the robot. There is also a button in the wristwatch user interface to remove the last functional waypoint in the trajectory.

### 3.2 Plan Visualization

Our interface was improved to provide a human-in-the-loop planning system. This planning system utilizes visualizations as well as alerts to inform the operator of the state of the plan. We also add a pre-check system to our interface to identify manipulation functional waypoints that are likely to fail and communicates this information to the operator. The operator can then adjust the plan on the fly, while they are setting up their trajectory, as opposed sending a trajectory that is likely to fail to the motion planner. Our pre-check system checks if manipulation functional waypoints are out of reach, meaning that they are further than the robot's arm's length from the robot. If a manipulation functional waypoint is

determined to be out of bounds by our pre-check system, then it is recolored orange to indicate a warning to the operator that it is very likely that the Maniuplation functional waypoint is unreachable by the robot. This method does not actually calculate a motion plan for the functional waypoints, it just removes some physically impossible to reach positions. Functional waypoints that are within the robots reach, but are unreachable by the motion planner due to impossible orientations can still lead to false positives. Our pre-check system will be further improved in future versions of our interface to prevent false positives.

If a manipulation functional waypoint is determined to be unreachable by the motion planner, the operator should be informed so that they can properly adjust their trajectory and send their new plan to the robot. To communicate this to the operator, Manipulationfunctional waypoints are recolored red, indicating that the robot encountered an error while planning to that point. A planner status message is also updated indicating in red text that the manipulation functional waypoint is unreachable. Figure 7 shows the state indication of the manipulation functional waypoint for a Fetch robot.

When the robot successfully plans the operator's trajectory, this plan should then be displayed to the operator so that they can make all necessary adjustments prior to approving the plan. To do this, we have a copy of the virtual robot execute the robot's planned trajectory. This enables the operator to visualize the robot's plan, with respect to the surrounding environment, before they approve for the actual robot to execute the trajectory. The operator can then adjust the robot's plan, if the robot's plan seems unsafe. This visualization method is similar to other commonly used visualization methods, such as the one proposed by Rosen et al. for augmented reality interfaces [16].

To provide human-in-the-loop planning, our interface must also communicate the state of the planner to the operator. The planner status is conveyed to operator with the following phrases:

- Ready to plan!
- Planning...
- Plan Successful!
- Executing Waypoint [#] / [Total #]
- Plan Failed at Waypoint [#]

## 4 NAVIGATION PLANNING IN VR

The previous iteration of our interface provided three methods for an operator to send navigation commands to a robot. First, the operator could press down the controller's trackpad to move the robot, a method very similar to controlling an RC car. The second method allowed the operator to create a navigation goal by clicking on a location in a minimap UI element, which would generate a goal at the corresponding point. The final method made use of a point and click method, where the operator would point to a location with the controller which is then sent to the robot as a navigation goal. In that iteration of our VR interface, the robot would then immediately create a navigation plan to the destination.

Since our prior implementation focused on a humanoid robot, footstep markers were generated to show the operator the robot's planned path. An operator could then move these footstep markers to adjust the robot's plan. While the path the robot would take

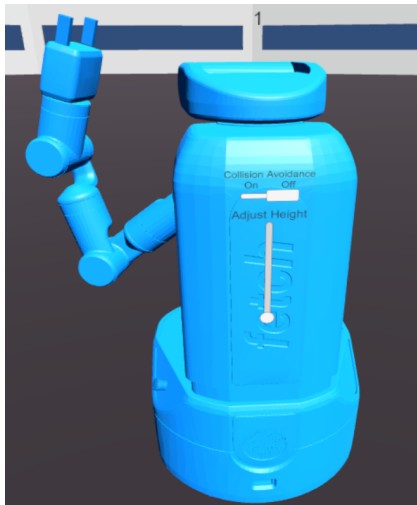

**Figure 8: Example of a navigation functional waypoint for the Fetch robot including the Fetch robot model, a label indicating the functional waypoint's order in the trajectory, a switch to enable or disable collision detection, and a slider interface to control the height of the robot.**

was displayed in the virtual world, the final position was not, so it could be difficult to tell exactly how close to an obstacle the robot would travel. There was also no way to send a series of waypoints that the robot should travel through. Additionally, when navigating, the operator may want to adjust the height of the robot at various points in its path, to get a better vantage point of the surrounding environment, which was not possible with our previous implementation.

The first improvement was creating new navigation functional waypoint that allows the operator to view the final goal state in the environment, preserving situational awareness. Much like the manipulation functional waypoint, a navigation functional waypoint consists of three visual components as seen in Figure 8.

To construct the navigation functional waypoints, we load the 3D model of the robot from the ROS robot description parameter, which allows the interface to be robot agnostic. By using a 3D model of robot, the operator is able to specify the position and orientation of navigation goals for the robot, and, most importantly, visualize them clearly in the context of the remote environment. Axes of these navigation goals are locked so that the goal can only be placed on the floor in a reachable orientation. Additionally, the robot model used as a navigation functional waypoint is updated to be in sync with the real robot. For example, if the operator creates a navigation functional waypoint and then moves the real robot's arms, the robot model in the functional waypoint will update to show the robot's new state. This ensures that the operator will understand how the robot will fit in its environment when it reaches the goal, which is important for preventing collisions.

These goals use the same labeling method as the manipulation functional waypoints. The numeric labels are anchored to the robot's head link, so if the robot changes its height, the label will move correspondingly and thus will always be visible.

Finally, the navigation functional waypoints have two UI elements anchored to its back. The first UI element enables the operator to toggle a collision avoidance mode on or off. When toggled on, an operator will not be able to place a goal which collides with a preloaded model in the environment. For example, with collision mode enabled, an operator can not place a Navigation Funcational Waypoint that would result in the robot colliding with the environment, such as a table. Collision avoidance can be turned off to accommodate scenarios where an operator may want to move a functional waypoint through a wall, in order to set a goal in a neighboring room. In these situations, an operator can disable collision avoidance, so that they can more conveniently move the goal through the wall rather than having to move all the way around the wall.

Below the collision avoidance toggle a vertical UI element enables the operator to adjust the height of the robot. Many mobile platforms have some means to adjust their height to get a better vantage point of the environment. For example, the Fetch robot has a torso that is driven by a motor. In this case the interface enables the operator to specify the height of the torso at this navigation functional waypoint. This interface is also applicable to humanoid robots, such as the Valkyrie R5, which can crouch down or stand up straight to get a better vantage point of their environment. Additionally, this interface can be extended to unmanned aerial vehicles (UAVs) where the slider value could be used to control the robot's height from the ground. When the slider interface for height control is moved and released by the operator, the navigation goal updates its robot model to the height specified by the operator. When executing the plan, the robot will first navigate through the path to reach the functional waypoint, then it will adjust its height to the operator specified height, if the operator moved the slider. After the robot has reached the desired height, the robot will then repeat the same process for each additional functional waypoint in the operator specified path.

## 4.1 Waypoint Creation and Modification

We attempted to keep the creation and modification of navigation functional waypoints as similar as possible to the manipulation functional waypoints previously discussed. We modified the point and click method to generate a navigation functional waypoint at the selected location instead of directly sending the goal to the robot. Just like manipulation functional waypoints, when the operator is holding a navigation functional waypoint, they can press the trigger button to place the current navigation functional waypoint and to create a brand new navigation functional waypoint, allowing the operator to easily create multiple navigation functional waypoints quickly. Like manipulation functional waypoints, the operator can create a navigation functional waypoint from within the navigation tab in the wristwatch user interface, which will spawn a navigation functional waypoint directly in front of the operator's virtual avatar. While more cumbersome than the other methods, we wanted to keep this as a fallback method and to allow the functional waypoints to have similar interaction methods independent of their functionality.

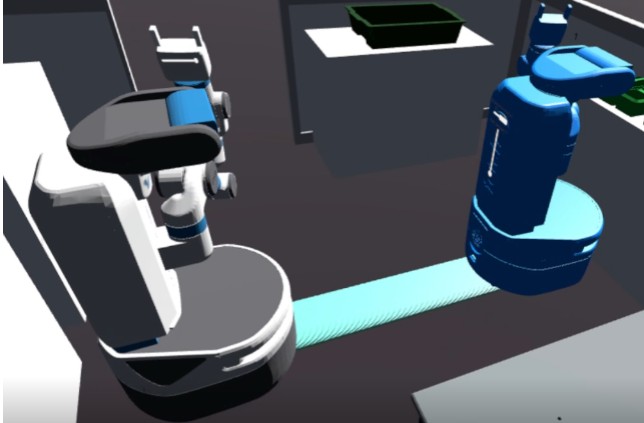

**Figure 9: The path visualization for a Fetch mobile robot.**

## 4.2 Path Visualization

As our previous work focused on humanoid robots, we would visualize the footsteps that the robot was planning to reach the navigation goal. We have expanded upon this to be more robot agnostic. In our new interface, we have added support for a non-humanoid mobile robot path type. The desired path type can be changed by selecting the mobility type of the robot from a drop down list in the Unity Editor. For non-humanoid mobile robots, the planned path will be shown with turquoise markers as seen in Figure 9.

Similarly to the manipulation functional waypoints, we have a pre-check for the navigation functional waypoints. If a navigation functional waypoint is colliding with a pre-loaded model of the environment, then it will be recolored orange, which indicates to the operator that the robot is likely unable to reach this goal. Finally, we display planner status messages using the same phrases as the manipulation functional waypoints.

## 5 ADDITIONAL CONTROLS

We found that there were several controls, such as adjusting the robot's height or gripper state, that we wanted to be able to use without using the planner. Thus, we added environment-anchored interface elements [17], which are anchored to either the robot or world coordinate systems, to provide control methods which were previously controlled exclusively within our wristwatch user interface.

When using mobile robots that can adjust their height, an operator may want to adjust the robot's height on the fly. To enable this, we attach a slider on the back of the virtual robot in a similar manner to the slider interface for height control on the navigation functional waypoints. This interface can be seen in Figure 10. When an operator moves the slider, a command will be sent immediately to the actual robot to adjust its height to the operator's desired height.

Additionally, an operator may want to open or close the robot's gripper on the fly, without using the manipulation functional waypoints. We have added a slider interface, just like the slider interface on the manipulation functional waypoint that enables the operator

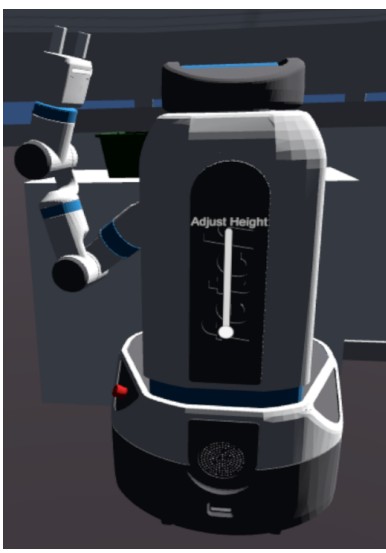

**Figure 10: The operator can control the height of the robot through the slider interface seen on the virtual robot's back, however, unlike the functional waypoint the robot's state is immediately updated.**

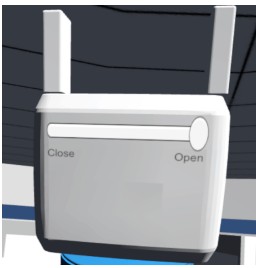

**Figure 11: The operator can control the state of the robot's gripper through the slider interface on the virtual robot's end effector, however, unlike the functional waypoint the robot's state is immediately updated.**

to control the robot's end effector without needing to use the planner or the wristwatch user interface. This interface can be seen in Figure 11. When an operator moves the slider, a command will be sent to the actual robot to open or close its gripper to the degree specified by the operator.

Finally, an operator might want the robot to look at a particular object or direction. This allows the operator to get a better understanding of the robot's environment through the point cloud. We have added an interface element where the operator can grab the virtual robot's head and pull down to where they want robot to look. When the operator grabs the robot's head, a 3D arrow is created and follows the operator's controller position as seen in Figure 12. This marker indicates where the robot will be looking once the operator releases the arrow.

We have also color coded our interface so that the operator can associate the colors with their corresponding control methods. All robot-related control methods use a light blue color, as seen in our

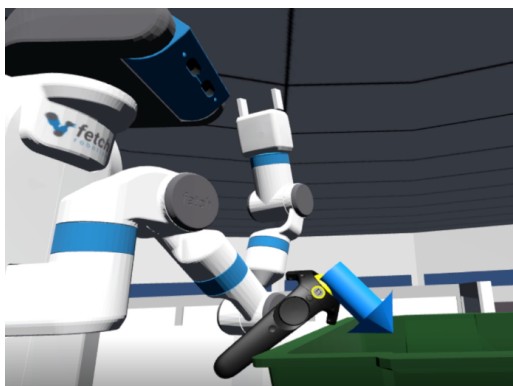

**Figure 12: The operator can command the robot to look at a particular point in the environment by moving the arrow marker to their desired location.**

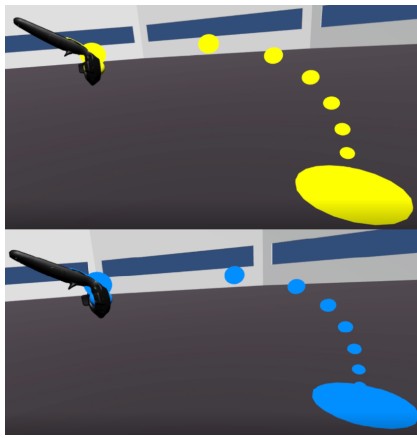

**Figure 13: This figure shows our color coded navigation control methods. The top image shows a yellow arc which points to the location that the operator will teleport to. The bottom image shows a blue arc from our point and click method, which shows the operator where a navigation functional waypoint will be generated.**

Manipulation and navigation functional waypoints. Controls for the operator are all color coded yellow. Figure 13 shows the color coded controller arcs for mobility controls. The yellow arc points to the location that the operator will teleport to when they release the trackpad button. Similarly, the blue arc is used for the navigation goal point and click method, after the operator releases the trackpad button, a navigation functional waypoint would be generated at the location they are pointing to.

## 6 CONCLUSION

In this paper, we have explained how our new interface expands upon our previous iterations by making the interface more robot agnostic, by enabling human-in-the-loop planning, and by introducing functional waypoints to allow the operator to command a remote robot in a way that preserves situation and task awareness.

We will investigate the effectiveness of this design in an upcoming user study using the Fetch robot.

## ACKNOWLEDGMENTS

This work has been supported in part by the National Science Foundation (IIS-1944584 and IIS-1451427), the Office of Naval Research (N00014-18-1-2503), the Department of Energy (DE-EM0004482), and NASA (NNX16AC48A).

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
