# OpenReview forum: "Semi-Autonomous Planning and Visualization in Virtual Reality"
_humanrobotinteraction.org/HRI/2021I/Workshop/VAM-HRI — VAM-HRI 2021 Oral_

### Official Review · AnonReviewer3 · 2021-03-03
**Funcional Waypoint Review**

**Rating:** 8
**Confidence:** 5

**Review:**

This paper discusses changes/improvements to a VR interface for robot control (mobile manipulator) and introduces the concept of Functional Waypoints which “differ from traditional waypoints, in that they not only provide intermediate goals for the desired trajectory but also enable the operator to send additional commands to alter the robot’s state at these intermediate points”. An example Functional Waypoint discussed in the paper is a Manipulation Functional Waypoint which allows for both position and gripper control via a slider within the waypoint target. Ways to create Functional Waypoints are also discussed to utilize the 3D nature of VR and help keep the operator oriented. The iterative improvements over the prior design are explained throughout the paper.

I found this paper informative and appreciate the many visuals included to aid the paper. The interface seems to have improved considerably.

 I think this paper would be stronger if the order of Functional Waypoints described were inverted. I found myself not seeing the difference between a waypoint with gripper control being different than the Manipulation Functional Waypoint. This can be found in work from DRC such as affordance templates (https://ieeexplore.ieee.org/document/7140073). The Navigational Functional Waypoints are a more clear identifier of state change ability that is both granular joint control as well as other state control (collision avoidance). I think the discrepancy with gripper control should be addressed to strengthen the idea of “Functional Waypoint”.

Nitpicks below:
Not sure “Functional Waypoint” should be capitalized as it is somewhat jarring to read/not sure it really should be a proper noun. Very much personal preference.

Inconsistent tense in System and Environment, typically these are reported in past tense.

For anything based in color, you may want to consider patterns instead or just be sure to reference colorblind friendly palettes (https://davidmathlogic.com/colorblind/#%23D81B60-%231E88E5-%23FFC107-%23004D40)

Paper worth looking into for reference: https://ieeexplore.ieee.org/document/7140073 (The Affordance Template ROS package for robot task programming)

Figure 10 and 11 seem to be repeats, I believe they are just representations of controls directly on the robot as opposed to waypoints. Likely these could be made smaller/side by side referencing the prior figures 3 and 8.

---

### Decision · Program_Chairs · 2021-03-06

Accept (Oral)